# Golgi compartments enable controlled biomolecular assembly using promiscuous enzymes

**Anjali Jaiman, Mukund Thattai***

Simons Centre for the Study of Living Machines, National Centre for Biological Sciences, Tata Institute of Fundamental Research, Bangalore, India

**Abstract** The synthesis of eukaryotic glycans – branched sugar oligomers attached to cell-surface proteins and lipids – is organized like a factory assembly line. Specific enzymes within successive compartments of the Golgi apparatus determine where new monomer building blocks are linked to the growing oligomer. These enzymes act promiscuously and stochastically, causing microheterogeneity (molecule-to-molecule variability) in the final oligomer products. However, this variability is tightly controlled: a given eukaryotic protein type is typically associated with a narrow, specific glycan oligomer profile. Here, we use ideas from the mathematical theory of self-assembly to enumerate the enzymatic causes of oligomer variability and show how to eliminate each cause. We rigorously demonstrate that cells can specifically synthesize a larger repertoire of glycan oligomers by partitioning promiscuous enzymes across multiple Golgi compartments. This places limits on biomolecular assembly: glycan microheterogeneity becomes unavoidable when the number of compartments is limited, or enzymes are excessively promiscuous.

## Introduction

The surfaces of all living cells are decorated with information-rich oligosaccharide molecules known as glycans: branched sugar oligomers covalently linked to proteins or lipids (*Varki et al., 2017*). Glycans encode cell identity and mediate a variety of intercellular interactions. They play critical roles in development, species recognition, self-nonself discrimination, and host-pathogen coevolution (*Varki, 2017*).

Glycans are composed of a small set of monosaccharide building block types (monomers) and disaccharide bond types (linkages in the branched glycan oligomer) (*Adibekian et al., 2011*; *Figure 1A*). Eukaryotic glycans are built by collections of glycosyltransferase (GTase) enzymes the ER and Golgi apparatus, a process known as glycosylation. GTase enzymes are chemically precise but contextually sloppy (*Taniguchi and Honke, 2014*; *Biswas and Thattai, 2020*): a given enzyme catalyzes a specific bond between a specific pair of monomer types, but can act promiscuously and stochastically on many oligomer types (*Figure 1B*). As a consequence, even a single cell with a limited set of GTase enzymes can theoretically synthesize an astronomical array of oligomeric combinations (*Cummings, 2009*).

A given glycosylated protein type in a given cell is usually associated with multiple glycan oligomers, a form of molecule-to-molecule variability known as microheterogeneity. The enormous potential for variability is exemplified by prokaryotic glycans, which are typically random heteropolymers (*Varki et al., 2017*, Chapter 21,22; *Adibekian et al., 2011*). In contrast, eukaryotic glycoproteins are typically associated with a narrow, specific set of glycan oligomers, referred to as protein's glycan profile (*Varki et al., 2017*, Chapter 1; *Campbell et al., 2014*). The low variability of eukaryotic glycans is functionally relevant: specific protein glycan profiles are associated with distinct species (*Hamako et al., 1993*; *Hård et al., 1992*; *Hokke et al., 1994*), distinct individuals (as with ABO

**\*For correspondence:**
thattai@ncbs.res.in

**Competing interests:** The authors declare that no competing interests exist.

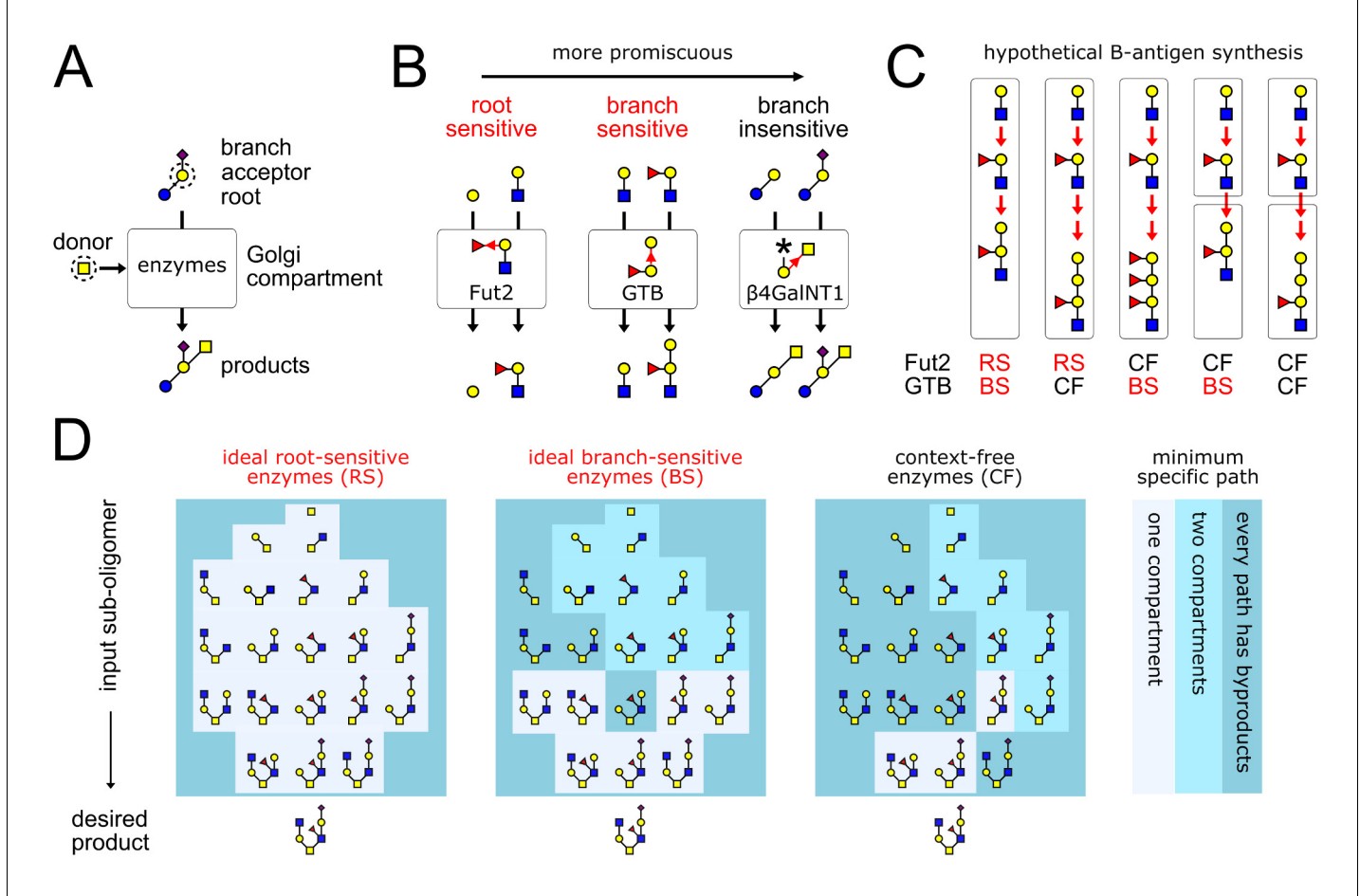

**Figure 1.** Glycan synthesis by promiscuous enzymes in Golgi compartments. (**A**) A GTase enzyme catalyzes a glycosidic linkage between a specific donor monomer type (the 'donor-substrate') and a specific acceptor monomer type with specific branches or roots (the 'acceptor-substrate'). We represent distinct monomer types in an oligomer by shapes/colors, and linkages between distinct monomer carbons by distinct bond angles (*Varki et al., 2017*). (**B**) GTase enzymes can show varying degrees of promiscuity, demonstrated here by three real examples of enzymes that act on galactose acceptors (*Taniguchi and Honke, 2014*, Chapter 39,44,47). We represent each GTase enzyme graphically, showing its acceptor-substrate and the specific monomer-addition reaction it catalyzes (red arrow from acceptor to donor at distinct angles for distinct acceptor carbons). The root-sensitive enzyme Fut2 requires the galactose acceptor to be linked to a GlcNAc root monomer. The branch-sensitive enzyme GTB requires its galactose acceptor to have a fucose branch. The branch-insensitive enzyme β4GalNT1 can act on branched or un-branched galactose monomers. (**C**) The localization of enzymes in successive Golgi compartments can sometimes mitigate the variability caused by enzyme promiscuity. This is demonstrated using the example of blood group B-antigen synthesis by the root-sensitive (RS) enzyme Fut2 and branch-sensitive (BS) enzyme GTB. We consider hypothetical scenarios in which these enzymes are context-free (CF). Increased enzyme promiscuity leads to tandem repeat byproducts. Placing the context-free Fut2 enzyme in a separate compartment from branch-sensitive GTB restores specific synthesis of the B-antigen. (**D**) Our main result is illustrated by this example (elaborated in *Figure 4*). We show the minimum number of compartments needed to specifically synthesize the same product oligomer from any of its sub-oligomers. We compare three broad classes of enzymes: ideal root-sensitive and branch-sensitive enzymes (ideal enzymes can read root chains or branches to arbitrary depth) or context-free enzymes. Ideal root-sensitive enzymes can distinguish the position of every acceptor monomer in an oligomer, so they can specifically synthesize any product in a single compartment. Branch-sensitive and context-free enzymes are more promiscuous; splitting such enzymes across multiple compartments increases the repertoire of oligomers they can specifically synthesize.

blood groups [*Patenaude et al., 2002*]) and distinct cell types in an individual [*West et al., 2010*]; and altered glycan profiles are implicated a variety of disorders (*Freeze and Ng, 2011*; *Gill et al., 2013*; *Lo-Guidice et al., 1994*).

How do eukaryotic cells generate narrow, specific protein glycan profiles despite the variability caused by promiscuous enzymes? Simulations suggest that the localization of enzymes within Golgi compartments plays a key role (*Liu et al., 2008*; *Spahn and Lewis, 2014*; *Spahn et al., 2016*; *Fisher et al., 2019*). Consider the GTases Fut2 and GTB, which specifically synthesize the blood group B-antigen (*Taniguchi and Honke, 2014*, Chapter 44,47). Hypothetical promiscuous versions

of these enzymes, if placed in a single compartment, generate variable tandem repeat oligomers as byproducts (*Figure 1C*); specific synthesis of the B-antigen is restored when the promiscuous enzymes are placed in two separate compartments. However, tandem repeat synthesis is just one among many sources of variability, and GTases can be more or less promiscuous than the enzymes in our example. We would like to understand the general conditions under which compartmentalization mitigates glycan variability.

Mechanisms for controlling variability are central to the field of algorithmic self-assembly, which explores how building blocks with sloppy interactions can be programmed to assemble into a desired final structure (*Soloveichik and Winfree, 2007*; *Zeravcic and Brenner, 2014*; *Murugan et al., 2015*). Here, we use ideas from self-assembly theory to precisely enumerate the possible sources of glycan variability. Using rigorous mathematical theorems, we demonstrate that compartmentalization mitigates variability for large classes of promiscuous enzymes (*Figure 1D*) (proofs of theorems are provided in Appendix 2). Since the number of distinct Golgi compartments in any cell is limited, microheterogeneity due to byproduct synthesis becomes unavoidable during the synthesis of complex oligomers. Nevertheless, multi-compartment synthesis greatly increases the set of oligomers eukaryotes can specifically synthesize (*Figure 1D*), complementing other mechanisms of enzymatic control such as kinetic regulation. Thus, a quintessential eukaryotic trait (intracellular compartments) provides the means to overcome a fundamental biochemical limitation (enzyme promiscuity). This capability may underlie the conservation of the Golgi across eukaryotes (*Barlow et al., 2018*): the idea of the Golgi apparatus as a factory assembly line is more than a metaphor, it is a mathematical and biological necessity.

## Results

### Glycan synthesis in Golgi compartments

We focus on the diverse class of O-glycans, which are associated with most eukaryotic cell-surface proteins (*Varki et al., 2017*, Chapter 10). The synthesis of O-glycan oligomers begins in the Golgi apparatus, when a root monomer is attached to a specific serine or threonine on a substrate protein. The Golgi consists of an ordered series of compositionally distinct compartments (for example, *cis*, *medial* and *trans* cisternae). Each compartment contains a specific set of GTase enzymes responsible for growing an oligomer on a specific protein type. Multiple models are proposed for how oligomers and enzymes are trafficked through the Golgi. In the transport model, oligomers are ferried through successive compartments; in the maturation model, oligomers remain in place while the enzymatic composition of the compartment undergoes a rapid switch-like transition between successive states (*Mani and Thattai, 2016*; *Pantazopoulou and Glick, 2019*). Under either scenario, whether transport or maturation, growing oligomers spend time within successive enzymatic compartment types. (The residence time of an oligomer within a compartment type is exponentially distributed in transport model, and sharply peaked in the maturation model; our results only depend on whether the average residence time $T$ is small or large compared to the average time for monomer addition.) The oligomer finally exits the last compartment in the series. The set of final oligomers associated with a given substrate protein makes up its glycan profile.

### GTase promiscuity and stochasticity

As the growing oligomer spends time within successive Golgi compartments, it encounters distinct collections of GTase enzymes (*Moremen et al., 2012*; *Pantazopoulou and Glick, 2019*). During each such encounter, the enzyme scans the oligomer for a site that matches a structural motif (the 'acceptor-substrate') and attaches a single free monomer (the 'donor-substrate') to that site (*Figure 1A*; *Liu et al., 2008*; *Moremen et al., 2012*). A given GTase enzyme invariably catalyzes a glycosidic (C-O-C) linkage between specific carbons on a specific free donor monomer type and a specific acceptor monomer type within the acceptor-substrate (*Figure 1A*). However, these enzymes are doubly sloppy: they are both promiscuous and stochastic.

#### Promiscuity

There are two strong reasons to expect GTase enzyme promiscuity. First, there are far fewer GTase enzymes than observed glycan oligomers in any given species (*Narimatsu et al., 2017*); moreover,

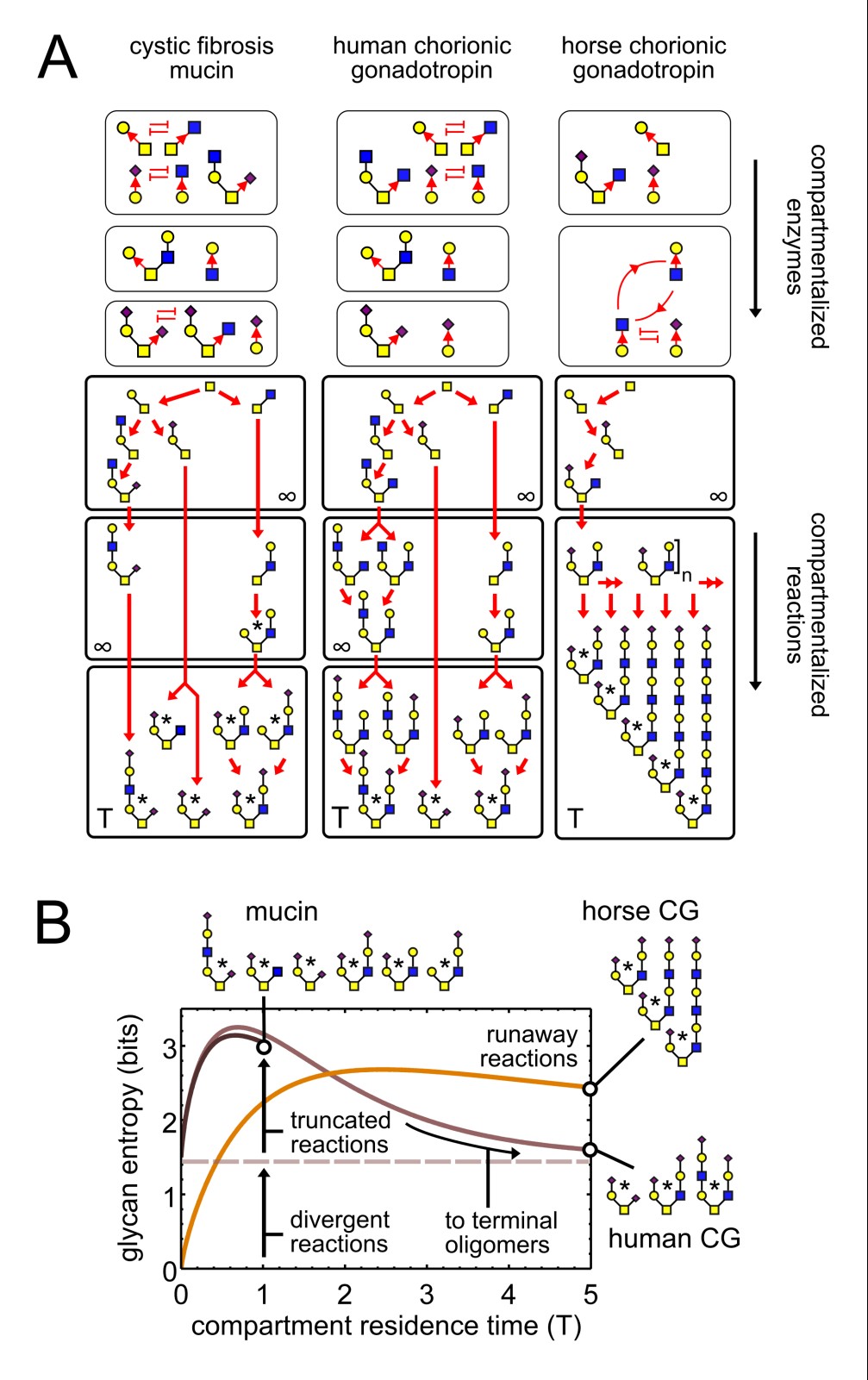

**Figure 2.** Observed patterns of glycan variability. (**A**) We compare oligomer profiles from three datasets (*Lo-Guidice et al., 1994*; *Hård et al., 1992*; *Hokke et al., 1994*); observed oligomers are starred, only non-fucosylated oligomers are shown. For each profile, we show a hypothetical set of compartmentalized enzymes leading to its synthesis; cyclic arrows represent runaway reactions, blunt arrows represent the action of one enzyme blocking the action of another. Each enzyme set is associated with a reaction network; red arrows show single-monomer-addition reactions. All

*Figure 2 continued on next page*

*Figure 2 continued*

oligomers exit the compartment as outputs after some average residence time. The residence time of the last compartment in each series is $T$; the residence time of all other compartments is $\infty$. (B) Effect of compartment residence time on glycan variability, quantified using Shannon entropy. The entropy of the compartment's output distribution depends on that of its input distribution, on the structure of the compartment's reaction network, and on its average residence time $T$.

the same highly conserved enzymes are found across species with very distinct glycans (*Kaneko et al., 2001*; *Figure 2*). Therefore, by the pigeonhole principle, an individual GTase enzyme must be promiscuous and able to act on many distinct oligomer types (*Narimatsu et al., 2017*). Second, although oligomers can be arbitrarily large, there is a limit to the size of the motif any enzyme can recognize within it. A given enzyme could act everywhere its recognition motif is found on an oligomer; the smaller the recognition motif, the more oligomers it will be found within, and the more promiscuous the enzyme (*Blixt et al., 2008*; *Taniguchi and Honke, 2014*; *Biswas and Thattai, 2020*). In our analysis we consider three broad classes of enzymes, exemplifying varying degrees of promiscuity (*Figure 1B*). Context-free enzymes are those whose acceptor-substrates are single monomers, no matter what branches or roots they are linked to. These enzymes are maximally promiscuous. Ideal branch-sensitive enzymes are those whose acceptor-substrates are some acceptor monomer type having or lacking specific branches. These are intermediate in promiscuity. Ideal root-sensitive enzymes are those whose acceptor-substrates are some acceptor monomer type linked to a specific root chain. These enzymes can distinguish every monomer in an oligomer and are minimally promiscuous. Enzymes that read roots and branches to partial depth represent more complex types of intermediate promiscuity; we do not discuss these possibilities here.

## Stochasticity

Chemical reactions within cells are necessarily stochastic, due to the low molecule numbers involved. While the growing oligomer is within a given compartment, the order in which it encounters the available GTase enzymes is equivalent to a Markov process of random sampling with replacement, with randomly distributed time intervals between successive encounters (*Gillespie, 1977*). The reaction network of the compartment shows every possible oligomer growth order starting from a given input oligomer, as a result of all possible enzyme-catalyzed single-monomer-addition reactions in all possible permutations (*Figure 2A*). Since O-glycan oligomers are not pruned, these reactions are irreversible (*Varki et al., 2017*, Chapter 6). Within a reaction network, intermediate oligomers are those that can potentially be further extended by some available GTase enzyme, and terminal oligomers are those that cannot be further extended. Two identical input oligomers might take different paths in the reaction network as they encounter GTase enzymes in different random permutations and at different times. An oligomer might encounter the same enzyme repeatedly (if the enzyme is at high concentrations), or it might exit the reaction compartment without ever encountering some enzyme (if the enzyme is at low concentrations).

## Patterns of glycan variability

Consider any assembly-line reaction in which oligomers are built by adding one monomer at a time. We can show that there are precisely three ways for variability to occur (Appendix 2, Remark 1): truncated, runaway, and divergent reactions. Each type of reaction variability corresponds to a specific output pattern that has been observed in real glycan datasets (*Figure 2*; *Table 1*; *Varki et al., 2017*, Chapter 1). These are exemplified by oligomer profiles of respiratory mucins from a cystic fibrosis patient (*Lo-Guidice et al., 1994*), human chorionic gonadotropin (CG) from a cell line

**Table 1.** Variability caused by promiscuous, stochastic enzymes.

| Observed product pattern | | Type of reaction variability | | Enzymatic cause |
|---|---|---|---|---|
| intermediate oligomers | $\Leftrightarrow$ | truncated reaction | $\Leftrightarrow$ | low concentration |
| tandem repeats | $\Leftrightarrow$ | runaway reaction | $\Leftrightarrow$ | linkage loop |
| mutually exclusive fates | $\Leftrightarrow$ | divergent reaction | $\Rightarrow$ | acceptor block |

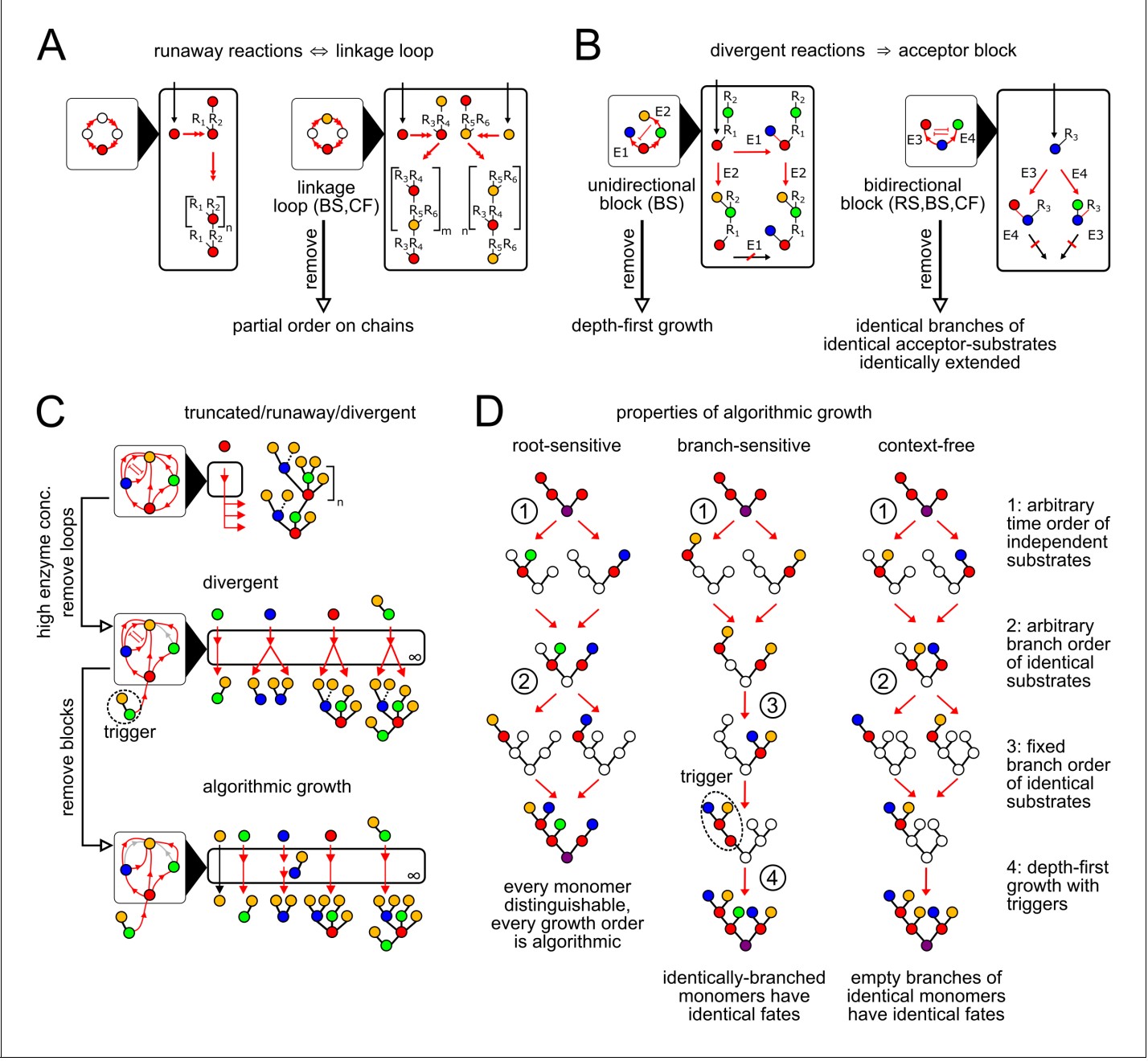

**Figure 3.** Enzymatic causes of glycan variability. (**A**) Runaway reactions occur whenever certain steps of oligomer growth can be iterated to produce tandem repeats. Loops in the linkage network are necessary and sufficient for runaway reactions (Appendix 2, Lemma 1). An acyclic linkage network imposes a partial order on monomer types along any chain. $R_i$ represents an arbitrary oligomer. Boxes with black triangles represent the linkage network, showing all orders in which monomer types can be linked, with arrows from acceptor monomer types to donor monomer types (Appendix 2, Remark 2). Double arrows in linkage and reaction networks represent multiple reaction steps. (**B**) Divergent reactions occur whenever the reaction network has a fork that can never reconverge. This occurs when the action of one enzyme blocks the subsequent action of another, so the fate of the oligomer depends on the random order of enzyme action. Blunt red arrows in the linkage network represent the action of one enzyme blocking the action of another. Acceptor blocks are necessary (but not sufficient) for divergent reactions (Appendix 2, Lemma 2). Unidirectional acceptor block (only branch-sensitive enzymes): the acceptor-substrate of one enzyme is on some branch of the acceptor-substrate of another. Bidirectional acceptor block (all enzyme classes): two enzymes compete for the same acceptor-substrate. (**C**) We start with a compartment containing an arbitrary collection of enzymes. We can eliminate truncated reactions by ensuring high enzyme concentrations. We can eliminate runaway reactions by removing (or disabling with triggers, for branch-sensitive enzymes) at least one enzyme involved in each linkage loop. We can eliminate divergent reactions by removing all but one enzyme involved in each block. The result is an algorithmic compartment: for each possible input, it specifically synthesizes a corresponding

*Figure 3 continued on next page*

*Figure 3 continued*

unique output. (D) Properties of algorithmic growth, for different enzyme classes. See *Appendix 3—figure 1* for a detailed example of branch-sensitive algorithmic growth.

(*Hård et al., 1992*), and horse chorionic gonadotropin (*Hokke et al., 1994*) (datasets from Uni-CarbKB [*Campbell et al., 2014*]). These patterns can be explained by the following enzymatic causes. For the mucins, two identical input oligomers exit the reaction compartment at different stages of growth; since some reactions are truncated, this gives a combination of both intermediate and terminal oligomers as outputs. For horse CG, a compartment contains two enzymes that drive a runaway reaction; this gives oligomers with an arbitrary number of tandem repeats. For both mucins and human CG, two enzymes compete for the same acceptor-substrate; this sets up a divergent reaction, with mutually exclusive oligomer fates depending on the random order of enzyme action.

We model stochastic assembly-line reactions as continuous-time Markov processes with constant transition probabilities per unit time. This provides a probability distribution over each possible fate of the final output oligomer. The Shannon entropy of this output distribution in bits captures the variability of the glycan profile; approximately, it is the log-base-two of the number of distinct high-abundance oligomers. Each source of variability makes a distinct contribution to the residence-time-dependent entropy, as seen for the hypothetical reaction networks in *Figure 2*. For horse CG, the input entropy of the final compartment is zero (since there is a unique input oligomer); for human CG and mucins the input entropy is 1.5 (since input oligomers are in a 1:1:2 ratio due to divergent reactions in earlier compartments). At short residence times, the entropy initially rises due to the exit of intermediate oligomers from truncated reactions. This corresponds to the mucin profile. At long residence times multiple intermediate oligomers converge to a few terminal oligomers, so the entropy decreases; this corresponds to the human CG profile. For the horse CG profile, the entropy stays high even at long residence times, due to synthesis of tandem repeats via runaway reactions.

## Enzymatic causes of glycan variability

The examples discussed above suggest that each type of glycan variability is connected to a distinct enzymatic cause. We now make this connection rigorous (*Table 1*).

A *truncated reaction* causes intermediate oligomers to be produced as outputs. This occurs whenever the average waiting time for monomer addition (a quantity inversely proportional to enzyme concentrations) is comparable to or greater than the compartment's residence time (Appendix 2, Remark 1). The only way all input oligomers are guaranteed to reach a terminal state (assuming no proofreading) is if enzyme concentrations are sufficiently high, or equivalently, the residence time $T$ is sufficiently long (schematically, $T \to \infty$).

A *runaway reaction* is an infinite path in the reaction network, giving oligomers with arbitrary numbers of tandem repeats as outputs. This implies at least one enzyme must act repeatedly along a single chain. Ideal root-sensitive enzymes don't permit runaway reactions, since their acceptor monomers are at a fixed depth from the root. To diagnose runaway reactions for branch-sensitive and context-free enzymes, we examine the compartment's linkage network, summarizing the allowed order of monomer linkages (*Figure 3A*; Appendix 2, Remark 2). A compartment contains a runaway reaction if and only if its linkage network contains one or more loops (Appendix 2, Lemma 1), since each loop corresponds to a tandem repeat. Branch-sensitive GTase enzymes can prevent linkage loops by using triggers (branched acceptor-substrates that cannot be synthesized within the compartment from a single monomer; *Figure 3D*).

A *divergent reaction* is a fork in the reaction network that never reconverges, with distinct paths leading to mutually exclusive oligomer fates as outputs. To diagnose divergent reactions, we examine the acceptor-substrates of every enzyme in the compartment. A fork occurs whenever distinct enzymes can act on the same oligomer to yield distinct products. If these enzymes could act in any order (for example, if they act on distinct empty acceptor monomers on the oligomer) then the reaction paths could reconverge after the fork. If a fork does not reconverge this implies that the action of one enzyme on an oligomer blocks the subsequent action of another (Appendix 2, Lemma 2). For root-sensitive and context-free enzymes, this can only occur if both enzymes compete to act on the same carbon of the same monomer (bidirectional acceptor block; *Figure 3B*). For branch-sensitive

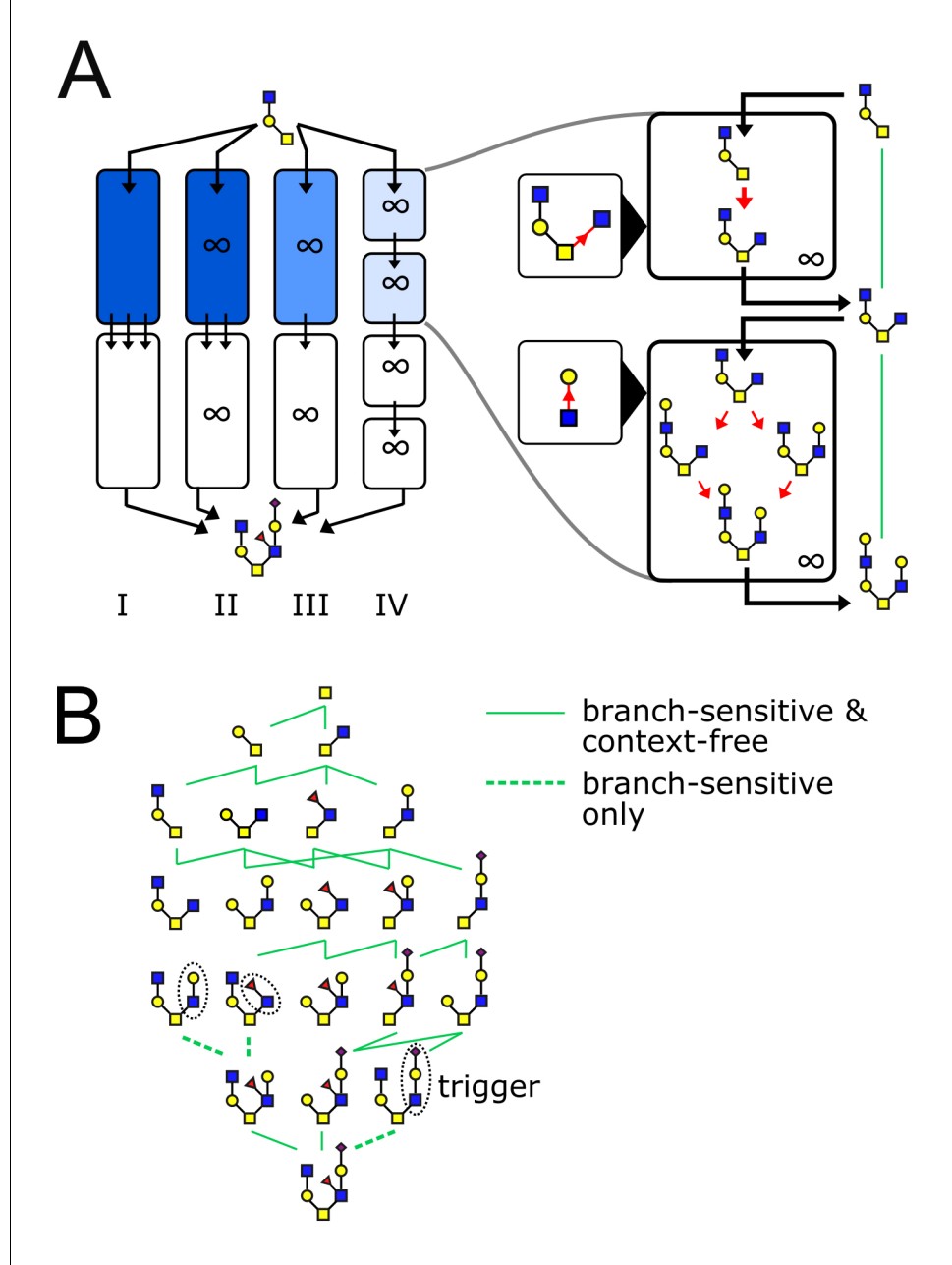

**Figure 4.** The glycan biosynthetic repertoire of multi-compartment systems. (A) Outline of Theorem 1 (Appendix 2). Suppose we are given a series of compartments that specifically synthesizes a desired target oligomer from a given input oligomer. We now proceed to modify the original compartments through several steps. Step I → II: Ensure enzyme concentrations are high, to eliminate intermediate oligomers. Step II → III: Remove all enzymes with acceptor blocks, to eliminate divergent reactions. Step III → IV: Replace each compartment with a series of infinite-residence-time compartments each containing a single enzyme. At each step, at least one of the original growth orders still remains, leading to the desired target oligomer. (B) We are given a target oligomer (bottom). We list all its sub-oligomers, and connect two oligomers by a green edge if some single-enzyme compartment can specifically synthesize the lower oligomer from the higher oligomer. All single-monomer additions can be achieved with ideal root-sensitive enzymes; only a subset of these can be achieved with branch-sensitive or context-free enzymes. To find the minimum number of compartments required to synthesize the target oligomer from any sub-oligomer, we decompose allowed paths into algorithmic growth stretches (*Figure 1D*).

enzymes, there are two ways this can occur: both enzymes have the same acceptor-substrate (bidirectional acceptor block; *Figure 3B*) or the acceptor-substrate of the first enzyme is on a branch of the acceptor-substrate of the second (unidirectional acceptor block; *Figure 3B*).

A compartment containing an arbitrary collection of enzymes might permit runaway and divergent reactions (*Figure 3C*, top). Our analysis shows that these sources of variability are not due to any individual enzyme: they arise from interactions within collections of enzymes (*Figure 3A,B*). We can eliminate truncated reactions by requiring high enzyme concentrations; we can eliminate linkage loops and acceptor blocks by removing certain enzymes from the reaction compartment (*Figure 3C*; Appendix 2, Lemma 3). These steps result in an 'algorithmic compartment': for each possible input, it specifically synthesizes a corresponding unique output. If we could watch an individual oligomer growing within an algorithmic compartment, we would find that its growth order always satisfied certain special properties that are easy to check, which we collectively characterize as 'algorithmic growth' (Appendix 2, Lemmas 4,5). The characteristics of algorithmic growth depend on the degree of enzyme promiscuity (*Figure 3D*). We now discuss how the concept of algorithmic growth can be used to find the minimum number of compartments required to specifically synthesize any desired oligomer.

## Controlled glycan synthesis in multiple compartments

Suppose we want to specifically synthesize a given target oligomer from a given input oligomer, with no byproducts. As a first attempt, we might pick an arbitrary growth order that leads, one monomer at a time, from input to target. Each monomer-addition reaction corresponds to the action of some enzyme. We could simply load a single compartment with this set of enzymes. This guarantees that the target oligomer will be synthesized from the given input. The problem is, various byproducts might also be synthesized due truncated, runaway, or divergent reactions; and the target oligomer might itself be further extended at long residence times. Since these problems arise due to interactions within collections of enzymes, they might be avoided by splitting the enzymes across several compartments. The following theorems provide the answer to two questions (*Figure 4A*; Appendix 2): Is specific synthesis of the target from the given input even possible? If so, what is the minimum number of compartments required?

### Theorem 1
A target oligomer can be specifically synthesized from an input oligomer if and only if it can be specifically synthesized from that input oligomer in a series of single-enzyme compartments.

### Theorem 2
A target oligomer can be specifically synthesized from an input oligomer in a series of $N$ compartments if and only if there is a growth order from the input to the target that can be fully decomposed into $N$ algorithmic growth stretches.

Theorem 1 provides an efficient protocol to search for a solution: we needn't consider all possible multi-enzyme combinations, it is sufficient to check single-enzyme compartments (*Figure 4B*). Once a solution is found using single-enzyme compartments, Theorem 2 provides a protocol to construct a solution using fewer multi-enzyme compartments. In this way, we are guaranteed to find the minimum number of compartments required for specific synthesis of the target (assuming a solution exists). For ideal root-sensitive enzymes a single compartment is always sufficient, since every growth order to a given terminal oligomer is algorithmic (Appendix 2, Lemma 4). The minimum-compartment path could be longer for the more promiscuous branch-sensitive or context-free enzymes (*Figures 1D* and *4B*). There are some situations in which glycan variability is unavoidable: certain oligomers can never be synthesized without byproducts, no matter how many compartments we allow. In *Appendix 3—figure 1* we discuss an example of a target oligomer that cannot be synthesized using branch-sensitive enzymes in fewer than two compartments.

## Discussion

Living systems excel at building complex structures using stochastic, unreliable molecular components. Macromolecules such as DNA, RNA and proteins are built by copying known target

templates, so errors can be removed by proofreading (*Murugan et al., 2012*). However, most biological structures are encoded using a step-by-step recipe – that is, an algorithm – rather than a template (*Navlakha and Bar-Joseph, 2011*). This is apparent during animal development: the genome encodes a recipe to make an adult, it is not a homunculus of the adult. Eukaryotic glycan synthesis is an exquisite demonstration of the same concept: re-configurable reaction networks act as template-free recipes to specifically encode diverse glycan oligomers.

In the absence of a template, cells use a spectrum of mechanisms to limit glycan variability. At one end are quantitative kinetic mechanisms (*Cardelli et al., 2018*), such as the regulation of compartment residence times (*Pantazopoulou and Glick, 2019*), control of donor-substrate levels (*Parker and Newstead, 2019*), or use of heteromeric enzymes to enhance sequential reactions (*Varki et al., 2017*, Chapter 4). At the other end are qualitative mechanisms that constrain which biosynthetic reactions are even allowed to occur: the selectivity of enzymes for their oligomer substrates, and the compartmentalization of enzymes within the Golgi. When enzymes are promiscuous, the kinetics of distinct reactions cannot be independently regulated, leading to variability (*Biswas and Thattai, 2020*). We have shown that, by splitting promiscuous enzymes across different Golgi compartments, this variability can be reduced or eliminated.

To synthesize oligomers without byproducts, cells must either decrease enzyme promiscuity, or increase the number of compartment types (*Figure 1D*). Both these strategies come at a cost, because there is a limit to the number of proteins a cell can encode. If all enzymes were highly selective, each with one specific substrate, a single compartment would suffice; but cells would require as many enzymes as oligomer types. On the other hand, an extensive system of membrane traffic proteins is required to maintain distinct Golgi compartments (*Pantazopoulou and Glick, 2019*). How do cells manage this trade-off? Some GTase enzymes, such as those which synthesize the N-glycan oligomannose precursor in the ER, are highly selective and appear to act on just one or two oligomers; but most GTases are promiscuous, able to act on many oligomers or at many points on a single oligomer (*Biswas and Thattai, 2020*). This means having multiple compartments is crucial for limiting glycan variability in real cells. These ideas inform strategies for artificial glycan synthesis (*Liu et al., 2019*) and algorithmic self-assembly (*Soloveichik and Winfree, 2007*; *Zeravcic and Brenner, 2014*; *Murugan et al., 2015*): 'stop-and-go' (*Liu et al., 2019*) or 'step-assembly' (*Doty, 2012*) approaches, which are analogous to multi-compartment synthesis, expand the repertoire of synthesizable oligomers.

The regulation of glycan synthesis by compartmentalization has biological advantages. By redistributing enzymes within the Golgi, multicellular organisms can use the same set of enzymes to generate distinct glycan profiles in distinct cell types (*Figure 2A*; *West et al., 2010*; *Fisher et al., 2019*). The rapid and reversible changes in glycan profiles seen during infection and inflammation (*Varki et al., 2017*, Chapter 46) are more consistent with changes in enzyme localization than changes in enzyme expression. Such changes could also facilitate rapid evolution of glycan profiles in the context of host-pathogen interactions and speciation (*Watanabe et al., 2020*; *Varki et al., 2017*, Chapter 20). But this also means small errors in Golgi localization could be pathological: glycan perturbations promoting tumor invasiveness appear to arise due to errors in enzyme localization, not mutations in the enzymes themselves (*Gill et al., 2013*; and congenital glycosylation disorders are often correlated with defects in Golgi structure (*Freeze and Ng, 2011*).

The emergence of intracellular compartments was a watershed step in eukaryotic evolution (*Dacks and Field, 2018*). Many hypotheses have been advanced about the adaptive function of such compartments. Here, we have shown that the compartmental organization of the Golgi apparatus allows cells to control glycan synthesis despite enzymatic promiscuity, potentially explaining why this remarkable organelle is universally conserved across all extant eukaryotic lineages (*Barlow et al., 2018*). The ability to generate narrow, specific glycan profiles would have been advantageous to early eukaryotes (*Wetzel et al., 2018*), enabling the sophisticated intercellular interactions that underlie sex, cooperation and multicellularity.

## Materials and methods

Methods are provided in the Appendices. Appendix 1: Definitions. Appendix 2: Proofs. Appendix 3: Detailed example of algorithmic growth.

## Acknowledgements

We thank Ajit Varki for introducing us to glycans, and Arnab Bhattacharyya for helping us view glycosylation through the algorithmic lens. We thank Ansuman Biswas, Ramya Purkanti, Somya Mani, Mugdha Sathe, Sachit Daniel, Kabir Husain and Amit Singh for useful discussions. We thank Kaadambari for critical inputs during writing.

## Additional information

### Funding

| Funder | Grant reference number | Author |
| --- | --- | --- |
| Simons Foundation | 287975 | Mukund Thattai |

The funders had no role in study design, data collection and interpretation, or the decision to submit the work for publication.

### Author contributions

Anjali Jaiman, Software, Formal analysis, Investigation, Visualization, Writing - original draft, Writing - review and editing; Mukund Thattai, Conceptualization, Formal analysis, Supervision, Funding acquisition, Investigation, Visualization, Writing - original draft, Writing - review and editing

### Author ORCIDs

Mukund Thattai (iD) https://orcid.org/0000-0002-2558-6517

### Decision letter and Author response

Decision letter https://doi.org/10.7554/eLife.49573.sa1
Author response https://doi.org/10.7554/eLife.49573.sa2

## Additional files

### Supplementary files

- Source code 1. MATLAB code to accompany *Figure 2B*.

- Transparent reporting form

### Data availability

Matlab source code has been provided for generating plots in Figure 2B.

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

# Appendix 1

## Definitions

### Glycan oligomer
A set of monomers linked to form a finite tree. Oligomers grow one monomer at a time, so every oligomer or sub-oligomer (any subtree of the oligomer) has a well-defined root monomer and a well-defined direction of growth (*Figure 1A*).

### Root chain
The path of specific linkages and monomers leading from the root to a given monomer (*Figure 1A*).

### Acceptor monomer
A monomer in an oligomer that can be linked to a new donor monomer at some specific carbon, through the action of some GTase enzyme (*Figure 1A*).

### Empty acceptor monomer
An acceptor monomer with nothing linked to any carbon, except the carbon on its root chain. A donor monomer becomes an empty acceptor monomer once it is linked to the oligomer (*Figure 1A*).

### Branch
The full sub-oligomer on a given carbon of a given monomer. If nothing is linked to a given carbon, we say the corresponding branch is empty. New branches are initiated when empty carbons are linked to donor monomers (*Figure 1A*).

### Tandem repeat
A chain in an oligomer that contains repeated instances of the same monomer type (*Figure 1C*).

### Acceptor-substrate
Root-sensitive enzyme: the acceptor-substrate is a specific acceptor monomer type with an empty branch at the carbon to be linked, and a specific root chain. Branch-sensitive enzyme: the acceptor-substrate is a specific acceptor monomer type with an empty branch at the carbon to be linked, and specific branches or empty branches at all other carbons. Context-free enzyme: the acceptor-substrate is a specific acceptor monomer type with an empty branch at the carbon to be linked (*Figure 1B*).

### Unidirectional acceptor block (branch-sensitive enzymes)
The acceptor-substrate of one enzyme is on a branch of the acceptor-substrate of another, so that the action of the first blocks the action of the second (*Figure 3B*).

### Bidirectional acceptor block (all classes of enzymes)

Two enzymes have the same acceptor-substrate, so the action of the first blocks the action of the second, and vice-versa. This is because the each enzyme can only act on the original unmodified acceptor-substrate (for example, both enzymes act on the same carbon of the same monomer) (*Figure 3B*).

### Compartment

A reaction compartment containing a set of specified GTase enzymes, and characterized by an average oligomer residence time (*Figure 1A*).

### Input oligomer

An oligomer or sub-oligomer that is provided as an input to a compartment (*Figure 1A*).

### Output oligomer

An oligomer that exits the compartment after some residence time, at some stage of growth (*Figure 1A*).

### Series of compartments

An ordered set of compartment types in which every output of each compartment type is passed as an input to the next compartment type. An initial input is provided to the first compartment, and the last compartment produces the final outputs (*Figure 1C*).

### Growth order

The order in which an oligomer is grown one monomer at a time, starting from a specific initial oligomer and leading to a specific final oligomer, through the action of successive enzymes in one or more compartments (*Figure 3D*).

### Depth-first growth order

A growth order in which, as soon as a new donor monomer is linked to an empty carbon of an acceptor monomer, the preexisting branches of that acceptor monomer no longer grow (*Figure 3D*).

### Reaction network

The nodes of a reaction network represent distinct oligomers, and its directed edges represent single-monomer-addition reactions. A reaction network shows all possible growth orders in a given compartment starting from a given input oligomer (*Figure 2A*).

### Terminal oligomer

An oligomer with no outgoing edges in the given reaction network (*Figure 2A*).

### Intermediate oligomer

An oligomer with at least one outgoing edge in the given reaction network (*Figure 2A*).

### Trigger

An acceptor-substrate that cannot be fully synthesized within a compartment starting from an empty acceptor monomer input (*Figure 3D*).

### Runaway reaction

A reaction network, starting from a given input oligomer, that has at least one infinite path (*Figure 2A*).

### Divergent reaction

A reaction network, starting from a given input oligomer, that has at least one fork beyond which reaction paths never reconverge (*Figure 2A*).

### Algorithmic compartment

A compartment that has no linkage loops or acceptor blocks. At infinite residence times, for each possible input it specifically synthesizes a corresponding unique output. Algorithmic growth is the type of growth that occurs in an algorithmic compartment (*Figure 3C*).

### Specifically synthesizable

A target oligomer is specifically synthesizable from an input oligomer if there is a series of one or more compartments that converts the input to the target as the unique final output, with no byproducts (*Figure 1D*).

## Proofs

### Remark 1. Variability in assembly-line reactions

Consider a reaction compartment in which monomers are added, one at a time, to a growing oligomer. Assume monomers are in excess, and that oligomers grow independently of one another. The oligomer starts in some input state, proceeds through a series of transitions, and exits the reaction compartment after some residence time $T$. We model this as a continuous-time Markov process over a discrete, potentially infinite state space: each state is an oligomer configuration; each transition is an enzyme-catalyzed single-monomer-addition reaction, whose probability per unit time is proportional to the corresponding enzyme concentration. These states and transitions form an acyclic assembly-line reaction network. Given a reaction network starting from some input oligomer, define a truncated network by cutting off every reaction path at an arbitrary point (this should correspond to an oligomer height much larger than the input oligomer height plus the number of monomer types). The terminal oligomers of the truncated network are then either terminal oligomers of the original network, or truncated oligomers containing arbitrary numbers of tandem repeats. For any finite residence time $T$ the exit probability for any oligomer in the truncated network is non-zero. As $T \to \infty$, or equivalently at high enzyme concentrations, the exit probability for any intermediate oligomer tends to zero, while the exit probability for any terminal oligomer or arbitrary tandem-repeat oligomer is non-zero. The input oligomer will be fully converted to a single specific final oligomer if and only if all the following conditions hold (*Table 1*): (a) The compartment has no runaway reactions so there are no arbitrary tandem repeat oligomers; (b) The compartment has no divergent reactions so every input leads to a unique terminal oligomer; (c) The compartment has a sufficiently long residence time (or equivalently, sufficiently high enzyme concentrations) so only this unique terminal oligomer exits as an output.

### Remark 2. Constructing a compartment's linkage network

Linkage networks are only defined for context-free and branch-sensitive enzymes; for these enzyme classes the fate of a bare monomer does not depend on its position in a tree. The nodes of a linkage network represent distinct monomer types or triggers; its directed edges represent acceptor-to-donor linkages at specific carbons. (a) To construct the linkage network of a compartment with context-free enzymes, we add an arrow from one monomer type to another if the corresponding acceptor-to-donor linkage can be carried out by some enzyme in the compartment. (b) To construct the linkage network of a compartment with ideal branch-sensitive enzymes is more involved. We first add an arrow from one monomer type to another if the corresponding acceptor-to-donor linkage occurs on any oligomer that can be synthesized starting from any empty acceptor monomer input. We need consider only oligomers whose height is less than or equal to that of the tallest acceptor-substrate of any GTase enzyme in the compartment. An enzyme whose acceptor-substrate is a trigger will not be represented among the arrows we have added so far. We must explicitly list each trigger and draw an arrow from its acceptor monomer to the relevant donor monomer type. A trigger is effectively a novel monomer type in the linkage network, such that other donor monomers can be added to it, but it cannot be added to other acceptor monomers. This gives the full linkage network of the compartment.

### Lemma 1

Runaway reactions ⇔ linkage loop.

## Proof

Consider a reaction network starting from some input oligomer. Keep all enzymes involved in this reaction network, remove other enzymes from the compartment. Suppose the reaction network has an infinite runaway path. Each reaction corresponds to the addition of one monomer to an oligomer. Therefore the reaction network contains at least one oligomer with an arbitrarily long root-to-tip chain. Since the number of monomer types is finite, the chain must include at least two instances of the same monomer type added within the compartment. Therefore the compartment's linkage network contains a loop. Conversely suppose a compartment's linkage network contains a loop. Then there is at least one monomer type, added at some step of the reaction network, on which a branch can be grown that includes another instance of the same monomer type. This process can be iterated ad infinitum to produce arbitrary tandem repeats. Therefore, the reaction network contains an infinite runaway path. (This argument is reminiscent of the pumping lemma for formal languages.) Runaway reactions are not possible with ideal root-sensitive enzymes, since they act at a defined depth from the root monomer (*Figure 3A*).

## Lemma 2

Divergent reactions $\Rightarrow$ acceptor block.

## Proof

Consider a reaction network starting from some input oligomer. Keep all enzymes involved in this reaction network, remove other enzymes from the compartment. Suppose the network has a divergent reaction. A fork in a reaction network occurs when two enzymes can act on the same oligomer. If the enzymes could act in either order the fork could immediately reconverge. Therefore, there is at least one pair of enzymes such that the action of the first enzyme blocks the subsequent action of the second. (a) For root-sensitive and context-free enzymes, there is only one way a divergent reaction can occur: both enzymes compete to act on the same carbon of the same monomer (bidirectional acceptor block). In this case, acceptor blocks are necessary and sufficient for a divergent reaction: there is no way for the reaction paths to reconverge. (b) For branch-sensitive enzymes, there are two ways a divergent reaction can occur: both enzymes have the same acceptor-substrate (bidirectional acceptor block) or the acceptor-substrate of the first enzyme is on a branch of the acceptor-substrate of the second (unidirectional acceptor block). In this case, acceptor blocks are not sufficient for a divergent reaction, as the reaction paths might reconverge via the action of a subsequent enzyme (*Figure 3B*).

## Lemma 3

If a compartment has a reaction network, starting from some input oligomer, that is finite (no linkage loops) with multiple terminal oligomers (due to acceptor blocks). Then there is an acceptor-block-free subset of the original enzymes that specifically synthesizes one of the original terminal oligomers from the input oligomer.

## Proof

Consider compartment with a finite reaction network starting from some input oligomer, with multiple terminal oligomers. Keep all enzymes involved in this reaction network, remove other enzymes from the compartment. Assume infinite residence time, so every possible growth order starting from the input oligomer reaches some terminal oligomer. Now there is at least one growth order in which a given enzyme involved in an acceptor block is completely blocked from acting because its acceptor-substrate or a branch of its acceptor-substrate is modified by some other enzyme. This growth order is retained when we remove the blocked enzyme. By iterating this procedure we eliminate every acceptor block. The resulting compartment is algorithmic (no linkage loops or acceptor blocks), so every growth order leads to just one of the original terminal oligomers (by Lemma 1 and Lemma 2). Removing blocked enzymes in a different order may select a different terminal oligomer; and not all terminal oligomers can necessarily be selected in this way.

## Lemma 4

All growth orders in an algorithmic compartment satisfy the following properties, which we collectively characterize as 'algorithmic growth'. (a) Context-free enzymes: Identical empty branches of identical monomers are identically extended. (b) Ideal branch-sensitive enzymes: Identical empty branches of identically branched monomers are identically extended; and growth is depth-first. (c) Ideal root-sensitive enzymes: Every growth order to a given terminal oligomer from any sub-oligomer is algorithmic.

### Proof

By definition, algorithmic compartments have no linkage loops or acceptor blocks. Therefore the reaction network starting from any input is finite, with a single terminal oligomer. Since there are no bidirectional acceptor blocks and the final oligomer is terminal, identical acceptor-substrates (defined separately for each class of enzymes) are identically extended at the same branch, and have identical fates in the final oligomer. (a) Context-free case: acceptor-substrates are some monomer type. (b) Branch-sensitive case: acceptor-substrates are some monomer type having or lacking specific branches. In addition, since there are no unidirectional acceptor blocks, once any enzyme acts on its acceptor-substrate its preexisting branches no longer grow. This implies growth is depth-first. (c) Root-sensitive case: acceptor-substrates are some monomer type with a specific root chain. Since every monomer in an oligomer has a distinct root chain, every growth order to a given terminal oligomer from any sub-oligomer is algorithmic.

## Lemma 5

A target oligomer can be specifically synthesized from an input oligomer in a single compartment if and only if there is an algorithmic growth order from the input to the target.

### Proof

Suppose a target oligomer can be specifically synthesized from an input oligomer in a single compartment. The reaction network starting from the input oligomer must have the target as its unique terminal oligomer, so there are no linkage loops. We can retain an acceptor-block-free subset of enzymes that specifically synthesizes this terminal oligomer from the input (Lemma 3). The resulting compartment is algorithmic, therefore any remaining growth order is algorithmic (Lemma 4). Conversely, suppose there is an algorithmic growth order starting from the input, with the target as the final oligomer. Each step of the growth order corresponds to the action of some GTase enzyme. Suppose we construct a compartment with just these enzymes. This will certainly synthesize the final oligomer from the input. However, we must check whether this set of enzymes generates runaway and divergent reactions that lead to the synthesis of tandem repeats or terminal oligomers other than the desired target. By assumption, identical empty branches on identical acceptor-substrates are identically extended, and therefore have identical fates in the final oligomer. This implies there are no bi-directional acceptor blocks. This also means no empty branch of the final oligomer can be extended by any enzyme, so it is terminal and there are no linkage loops. In the branch-sensitive case growth is depth-first, meaning no branch of the acceptor-substrate of one enzyme is later extended by another enzyme, so there are no unidirectional acceptor blocks. Therefore, the constructed compartment is algorithmic. At infinite residence times, it will specifically synthesize the target oligomer from the input oligomer.

## Theorem 1

A target oligomer can be specifically synthesized from an input oligomer if and only if it can be specifically synthesized from that input oligomer in a series of single-enzyme compartments.

## Theorem 2

A target oligomer can be specifically synthesized from an input oligomer in a series of $N$ compartments if and only if there is a growth order from the input to the target that can be fully decomposed into $N$ algorithmic growth stretches.

### Proof

We prove both theorems together. Suppose the target oligomer can be specifically synthesized from an input oligomer in a series of $N$ compartments (Stage I, *Figure 4A*). The outputs of each compartment are passed as inputs to the next compartment. Every possible growth order starting from the initial input oligomer leads to the target as the unique final oligomer, so we know there are no arbitrary tandem repeats generated in any compartment, and therefore no linkage loops. A subset of these growth orders will remain if each compartment has infinite residence time (Stage II). Each reaction network starting from each input of each compartment is finite. Therefore, we can replace the enzymes in each compartment with an acceptor-block-free subset that specifically synthesizes one of the original terminal outputs from any of the original inputs (Lemma 3). This leaves a set of $N$ algorithmic compartments (Stage III). All remaining growth orders start from the initial oligomer, pass through just one terminal oligomer of each successive compartment, and produces the target oligomer as the unique terminal output of the last compartment. Theorem 1: Among the growth orders at Stage III, since enzymes act in every possible permutation, there is at least one growth order in which each given enzyme acts successively on every available instance of its acceptor-substrate on the oligomer, before the next enzyme acts. This growth order is retained even once we replace each compartment by a series of single-enzyme infinite-residence-time compartments (Stage IV). The converse is trivial. Theorem 2: Since every compartment is algorithmic at Stage III, any growth order within each compartment is algorithmic (Lemma 4). Conversely, suppose there is a growth order from the input to the target that can be fully decomposed into $N$ algorithmic stretches. Then each stretch can be achieved within a single compartment (Lemma 5) so the target oligomer can be specifically synthesized from an input oligomer in a series of $N$ compartments.

## Appendix 3

### Detailed example of algorithmic growth

We want to test (*Appendix 3—figure 1A*) if a desired target oligomer (full structure, filled and empty circles) can be specifically synthesized from a given input oligomer (filled circles only) in one compartment. This is possible if and only if there exists an algorithmic growth order from the input to the target (Appendix 2, Theorem 2). We can already see that identical empty branches of different instances of monomer type $A$ in the input oligomer (red) have distinct fates in the target oligomer, so algorithmic growth is not possible using context-free enzymes. So we look instead for a branch-sensitive solution.

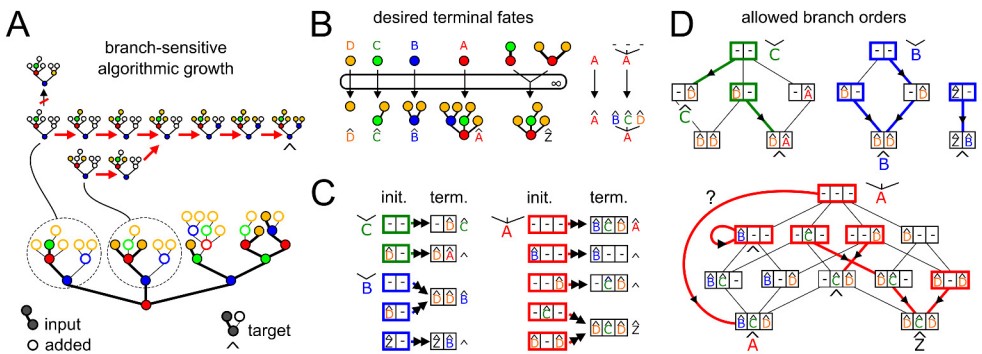

**Appendix 3—figure 1.** Efficient search for a branch-sensitive algorithmic growth order from input to target.

We show examples of possible growth orders for two sub-oligomers (*Appendix 3—figure 1A*, top). At each growth step an empty circle becomes filled as a new monomer is added. Depth-first growth means no new branch can be initiated on a monomer with incomplete existing branches; the leftmost vertical arrow is not a depth-first step. Identical acceptor-substrates must be identically extended at the same branch; therefore if two growth paths converge to the same acceptor-substrate, they must have the same subsequent growth step.

The input oligomer contains many distinct acceptor-substrates, each potentially in multiple copies, and each with a desired terminal fate (labeled with a $\Lambda$). A few examples of these are shown here (*Appendix 3—figure 1B*). The terminal fates of empty monomers ($\hat{A}$, $\hat{B}$, $\hat{C}$, $\hat{D}$) are of particular interest, since these are the allowed fates of any newly-initiated branches. Under depth-first growth it is sufficient to consider how new branches are initiated on acceptor-substrates whose existing branches are already terminally extended. Instead of a graphical representation (*Appendix 3—figure 1B*), we use a recursive representation showing the branches linked to each carbon of each acceptor-monomer type (*Appendix 3—figure 1C*). Thus, oligomer $\hat{A}$ is monomer $A$ linked to branches $\hat{B}$, $\hat{C}$, $\hat{D}$ on its three carbons. We list the set of distinct initial acceptor-substrates (bold boxes colored by acceptor monomer type) and their desired terminal fates in the target oligomer ($\Lambda$). Boxes are colored according to acceptor monomer type; each slot shows any existing terminally-extended branches; empty carbons are labeled '$-$'. We have not shown the trivial case of monomer type $D$.

For branch-sensitive enzymes, an algorithmic growth order is essentially determined by the choice of branch order. (Depth-first growth is enforced by assumption, since we consider only acceptor-substrates with terminally-extended existing branches.) We must find a single consistent branch order for each monomer type, such that each distinct acceptor-substrate achieves its desired terminal fate; we need consider only branches that are actually observed in the target oligomer. The initiation of all possible branches in all possible orders is represented as a transition graph (*Appendix 3—figure 1D*). Each node of the graph represents distinct acceptor-substrates, using the box notation from *Appendix 3—figure 1C*. Each directed edge represents the initiation and terminal extension of a branch on an empty

carbon. Bold colored arrows show a possible choice of successive branch additions, from initial acceptor-substrates (bold boxes) to desired terminal fates ($\Lambda$). There can be no bold outward arrows from terminal fates; the bold self-loop represents an acceptor-substrate that is already at its desired terminal fate. By the definition of algorithmic growth, there can be only one bold outward arrow from each acceptor-substrate. Any acceptor-substrate not reachable from an empty monomer is a trigger. The branch order search might have a single unique solution, as with monomer type $C$; or multiple solutions, as with monomer type $B$ (only one solution is shown here). There may be no solutions, since each choice of branch initiation cuts off other paths. In this example, there is no set of bold arrows that simultaneously achieves all desired terminal fates for monomer type $A$. For the choice of arrows shown here, all desired terminal fates are achieved except for that of the empty monomer $A$. Therefore, the given target cannot be specifically synthesized from the given input in a single compartment using branch-sensitive enzymes. We leave it as an exercise for the reader to show that the target oligomer can be specifically synthesized from the input oligomer by branch-sensitive enzymes in two compartments.

