## [Decision Letter]

**Acceptance summary:**

The manuscript addresses the interesting question of how, despite the sloppiness and promiscuity of the glycoenzymes, a limited variety of glycans are produced in the Golgi. The authors develop a complete formal theoretical framework and show that the sub-Golgi compartmentalization is key. It allows distributing the enzymatic activities through multiple processive Golgi compartments which effectively reduces the variability in the glycan output.

**Decision letter after peer review:**

Thank you for submitting your article "Golgi compartments enable controlled biomolecular assembly using sloppy enzymes" for consideration by *eLife*. Your article has been reviewed by three peer reviewers, and the evaluation has been overseen by a Reviewing Editor and Naama Barkai as the Senior Editor.

The reviewers have discussed the reviews with one another and the Reviewing Editor has drafted this decision to help you prepare a revised submission.

Summary:

The manuscript addresses the question of the control of glycan variability in the Golgi achieved by sloppy glyco-enzymes from a theoretical perspective. The authors propose that the variability issue can be solved by a spatial segregation of the enzymes, which could a posteriori provide an explanation for the functional origin of the Golgi compartmentalization.

Essential revisions:

The three reviewers and myself think that your work could be of strong interest for biologists working on membrane traffic and specially on Golgi, but in its current form, the message and the predictions for practical tests on living cells are not easily accessible to the majority of *eLife* readers. Therefore, we consider that you should deeply revise your manuscript to address this serious concern and discuss a few practical cases. In addition, since the dynamical structure of the Golgi is still a matter of debate, we think that you should add a discussion on how your model depends on Golgi maturation, on the consequence of a continuous gradient vs discrete distribution of the enzymes, on the impact in a multiple-compartments vs. a single compartment system. Eventually, we would like that you discuss the robustness of the model to the "uniform depth-first growth" rule.

[Editors' note: further revisions were suggested prior to acceptance, as described below.]

Thank you for submitting your article "Golgi compartments enable controlled biomolecular assembly using promiscuous enzymes" for consideration by *eLife*. Your article has been reviewed by three peer reviewers, and the evaluation has been overseen by a Reviewing Editor and Naama Barkai as the Senior Editor. The reviewers have opted to remain anonymous.

The reviewers have discussed the reviews with one another and the Reviewing Editor has drafted this decision to help you prepare a revised submission.

Summary:

This manuscript addresses theoretically the question of the control of glycan variability in the Golgi achieved by sloppy glyco-enzymes. The authors propose that the variability issue can be solved by a spatial segregation of the enzymes, which could a posteriori provide some functional origin to the Golgi compartmentalization.

Essential revisions:

The reviewers have appreciated your effort to facilitate the accessibility of your work to biologists reading *eLife* articles in your revised version. Nevertheless, we think that your paper could have a much stronger impact among biologists if you could further modify the presentation. We recommend following suggestions from reviewer #2 (see below).

Reviewer #2:

What I like about the work is the element of bringing in a new dimension into the intricate biological processes in the Golgi that involve glycan synthesis, binding, and self-assembly. However, I find the choice of language in the manuscript seriously concerning. The authors have chosen to present their main idea as mathematical theorems supported by proofs and lemmata. If the aim for this choice were to give the readers the assurance that what they are seeing is unquestionable fact and not just the opinion or suggestion of theoreticians, I believe this could be done in a much more accessible way. If the aim were to purport that the authors are making a new contribution to the field of mathematical logic by proving previously unknown theorems, then the aim would be incorrect and inappropriate. This has already been done in the specific context of algorithmic self-assembly in the papers that the authors cite themselves. The authors are merely applying these ideas to the specific biological problem at hand.

The main tenet of the authors' argument is enumerating the number of mutually exclusive and complementary possibilities in polymerization. This is basically equivalent to saying that a coin has two sides. One can, of course, make this into a theorem in mathematical logic and provide a proof of it, which can take pages and pages. However, what a normal user of this theorem would want to know from it will only be that we have only two mutually exclusive possibilities when we are throwing a coin: heads and tails. If we throw a dice, we have six possibilities. That's it. I do not see why the authors have chosen the rather heavy mathematical language to motivate the use of this simple enumeration of possibilities, for this particular biological application.

In response to my previous recommendation to streamline the presentation and make it more accessible to the appropriate biological readership of the journal, the authors moved some of the theorems and proofs to the appendix. As a result, the main manuscript now starts from lemma 3 and then wanders into some other mathematical blocks along the way. The only biological analysis is done in the captions of the figures, which are excessively long. Figure 1's caption is 29 lines if I managed to count correctly.

I think as it stands the manuscript will likely not attract its full potential in readership because of this confused style. My suggestion to the authors is to remove the mathematical language altogether and write the manuscript as if they are applying the enumeration argument to the biological process. They can motivate it, provide the examples next, and conclude by arguing for the generality of the principles and the arguments. I imagine this would make a wonderful paper that is also going to be read by the readers of *eLife*.

I cannot recommend publication of the manuscript in its current style, not even with the revision.

---

## [Author Response]

Essential revisions:The three reviewers and myself think that your work could be of strong interest for biologists working on membrane traffic and specially on Golgi, but in its current form, the message and the predictions for practical tests on living cells are not easily accessible to the majority of eLife readers. Therefore, we consider that you should deeply revise your manuscript to address this serious concern and discuss a few practical cases.

We have taken very seriously the input that our work should be more accessible to biologists. Before preparing our revision, we circulated the manuscript to cell biology colleagues to solicit specific feedback on (a) relevance of results and (b) clarity of exposition. Based on this feedback and on the reviewer comments, we have made major revisions to the paper, summarized here:

New emphasis: We received feedback that our analysis of promiscuity is potentially the most useful direct result of our work. We have therefore revised the manuscript to focus deeply on the causes and consequences of enzyme promiscuity. Both in the title and throughout the text, we have replaced the word “sloppy” (which was borrowed from the self-assembly literature) with the word “promiscuity” which is more informative and relevant to biologists. We have added several paragraphs to the Discussion, in which we outline the trade-off between promiscuity and compartmentalization.

New results: We have added completely new results, which compare the behavior of two classes of enzymes that span a range of possible promiscuity: the maximally promiscuous “branch-insensitive enzymes” and the less promiscuous “branch-sensitive enzymes”. This includes two new Lemmas (3 and 4) in Supporting Information. Our discussion is updated to show how promiscuity and compartmentalization jointly influence glycan variability, with several new predictions and new examples.

Modified figures: We have modified all figures (Figures 1, 2, 3 and Appendix 3—figure 1) to update our comparison of branch-insensitive and branch-sensitive enzymes. Our key new results are summarized in an example discussed in Figure 3C.

Streamlined text and terminology: We have replaced several technical terms with more informative terms and added multiple paragraphs that provide a non-technical background to the mathematical theorems. We have replaced the term “uniform depth-first growth” with “algorithmic growth” and expanded its definition to apply to both branch-insensitive and branch-sensitive enzymes (Lemma 4).

In addition, since the dynamical structure of the Golgi is still a matter of debate, we think that you should add a discussion on how your model depends on Golgi maturation, on the consequence of a continuous gradient vs discrete distribution of the enzymes, on the impact in a multiple-compartments vs. a single compartment system.

We thank the reviewers for pointing out this lacuna in our discussion. We have added new text (in the section “Glycan synthesis by promiscuous, stochastic enzymes”) that discusses in detail the implications of the transport and maturation models of the Golgi. We discuss why, as long as the maturation transitions are sufficiently rapid, our results will apply to both models. We have noted that our results will not apply to the case of a gradient of enzymatic compositions.

Eventually, we would like that you discuss the robustness of the model to the "uniform depth-first growth" rule.

We thank the reviewers for this insightful comment. Indeed, the definition of “uniform depth-first growth” is not universal but depends on the nature of the underlying enzyme promiscuity. In our revision we have explicitly derived independent results for two classes of enzymes that span a range of promiscuity: branch-insensitive and branch-sensitive enzymes. We have replaced the term “uniform depth-first growth” (which applies only to the latter class) with the broader term “algorithmic growth” and described the different types of algorithmic growth which apply to the different enzyme classes (Lemma 4). Our main result (Theorem 2) still applies, to both classes of enzyme promiscuity. Since real GTase enzymes will have intermediate promiscuity between the two classes, our results therefore provide both an upper and a lower bound on the number of compartments needed to specifically synthesize a given oligomer (Figure 3C). We therefore expect our predictions to be applicable in many cellular contexts.

[Editors' note: further revisions were suggested prior to acceptance, as described below.]

Essential revisions:[…]Reviewer #2:What I like about the work is the element of bringing in a new dimension into the intricate biological processes in the Golgi that involve glycan synthesis, binding, and self-assembly. However, I find the choice of language in the manuscript seriously concerning.

We thank the reviewer for this suggestion. We have extensively revised the figures and text of the manuscript. We have streamlined the flow of text to focus on just the key ideas and remove distracting technical asides.

The authors have chosen to present their main idea as mathematical theorems supported by proofs and lemmata. If the aim for this choice were to give the readers the assurance that what they are seeing is unquestionable fact and not just the opinion or suggestion of theoreticians, I believe this could be done in a much more accessible way.

This is an important point. We had chosen to present our results in the form of theorems, rather than the alternative possibility of simulations. Our aim was three-fold. First, we hoped to highlight general lessons, which is extremely difficult using simulations since any implementation of a simulation involves making several specific choices. Second, we wanted to ensure that all our assumptions were clearly stated, and that we did not miss any hidden assumptions about the behavior of enzymes. Third, we sought to provide a bridge to self-assembly researchers, who we hoped would be stimulated by ideas derived from this compelling biological example of controlled synthesis. We hope our updated text, involving more examples to buttress the mathematics, is less distracting and provides a more accessible exposition of the central ideas.

If the aim were to purport that the authors are making a new contribution to the field of mathematical logic by proving previously unknown theorems, then the aim would be incorrect and inappropriate. This has already been done in the specific context of algorithmic self-assembly in the papers that the authors cite themselves. The authors are merely applying these ideas to the specific biological problem at hand.

Although the self-assembly literature has a rich body of mathematical theorems, to our knowledge there is no specific application of these ideas to the construction of tree-like objects, which glycans are an example of. The basic idea of variability (runaway and divergent reactions) has been previously explored. However, the theorems we have proved go beyond application of existing theorems, and to our knowledge they are not to be found elsewhere the literature.

The main tenet of the authors' argument is enumerating the number of mutually exclusive and complementary possibilities in polymerization. This is basically equivalent to saying that a coin has two sides. One can, of course, make this into a theorem in mathematical logic and provide a proof of it, which can take pages and pages. However, what a normal user of this theorem would want to know from it will only be that we have only two mutually exclusive possibilities when we are throwing a coin: heads and tails. If we throw a dice, we have six possibilities. That's it. I do not see why the authors have chosen the rather heavy mathematical language to motivate the use of this simple enumeration of possibilities, for this particular biological application.

One of our major results is, indeed, the enumeration of mutually exclusive possibilities. The reviewer is correct that the application of this result does not require the detailed machinery of the theorem. However, one of our key contributions (summarized Table 1) is to show that the possibilities we have listed are indeed comprehensive. To prove we have not missed out other sources of variability, we required the full logical machinery. However, we agree that these details need not be in the main text. We have restricted them to the Appendices.

In response to my previous recommendation to streamline the presentation and make it more accessible to the appropriate biological readership of the journal, the authors moved some of the theorems and proofs to the appendix. As a result, the main manuscript now starts from lemma 3 and then wanders into some other mathematical blocks along the way. The only biological analysis is done in the captions of the figures, which are excessively long. Figure 1's caption is 29 lines if I managed to count correctly.

We have made major changes to the flow of the text, with the inclusion of new examples, new figure panels, and re-ordering of key results. We have removed the distracting technical excursions, instead we now motivate the result with several examples, before moving on to their mathematical basis. We agree the figure captions were too long in the previous version. We have re-organized the text, so the captions contain only details needed to interpret the figures, while the text contains fuller discussion and examples.

I think as it stands the manuscript will likely not attract its full potential in readership because of this confused style. My suggestion to the authors is to remove the mathematical language altogether and write the manuscript as if they are applying the enumeration argument to the biological process. They can motivate it, provide the examples next, and conclude by arguing for the generality of the principles and the arguments. I imagine this would make a wonderful paper that is also going to be read by the readers of eLife.

We thank the reviewer for taking this broad view of the manuscript. We have now done exactly this, started with examples, provided some intuition for them, and concluded with arguments for generality. All the material needed to see what’s “under the hood” is still contained in the Appendices for the technical reader, and we have directed readers to the relevant Lemma or Theorem at appropriate points in the text.